# Effect of Heat Treatment on Creep Deformation and Fracture Properties for a Coarse-Grained Inconel 718 Manufactured by Directed Energy Deposition

**DOI:** 10.3390/ma16041377

**Published:** 2023-02-06

**Authors:** Ying Li, Pavel Podaný, Martina Koukolíková, Jan Džugan, Tomáš Krajňák, Jozef Veselý, Srinivasan Raghavan

**Affiliations:** 1COMTES FHT a.s., Prumyslova 995, 334 41 Dobrany, Czech Republic; 2Department of Physics of Materials, Faculty of Mathematics and Physics, Charles University, 121 16 Prague 2, Czech Republic; 3Makino Asia Pte Ltd., 2 Gul Ave., Singapore 629649, Singapore

**Keywords:** laser directed energy deposition, Inconel 718, creep, dislocation cells, fracture mechanism

## Abstract

The creep properties of a laser-directed energy deposition (L-DED) technique manufactured Inconel 718 (IN718) was investigated at 650 °C/700 MPa. Microstructure and creep properties of L-DED IN718 samples were tailored by various post heat treatments involving homogenization heat treatment with temperature ranging from 1080 to 1180 °C + double aging and hot isostatic pressing (HIP). Microstructural changes and their influence on the creep behavior and fracture mechanism were observed and discussed. The results show that L-DED sample heat treated by a simple double aging exhibits a 49% increase in creep lifetime t_r_ and a comparable creep elongation ɛ_f_ when compared to the wrought material, due to the reserved coarse dislocation cell substructure from the L-DED process. The loss of dislocation cell structure and the coarsening of grains at higher temperature of heat treatments contributes to a shorter t_r,_ ε_f_, but faster ε˙_min_ (minimum creep rate). The present work demonstrates that a simultaneous improvement of creep strength and creep elongation can be achieved in the case of a coarse-grained L-DED IN718 by a double aging treatment which can preserve both the strengthening precipitates and an appropriate size of dislocation cells.

## 1. Introduction

Directed Energy Deposition (DED) is a popular additive manufacturing (AM) process in which a focused energy source is used to melt materials in the form of powder or wire as they are being deposited. This new AM technique has shown great potentials in wide applications, particularly the fabrication of functionally graded materials (FGMs) [1,2]. Manufacturing AM-FGMs through DED has spread to many metallic system in recent years. Among these DED-FGMs, gradients from steel-based alloys to nickel-based alloys, i.e., FGMs 316L SS-IN718 has attracted extensive attention for many applications in which superior strength and corrosion resistance at elevated temperature is required, e.g., nuclear power generation and oil refineries [3,4,5,6,7]. This FGMs system, however, still faces some major challenges before being used in those specific applications in which creep damage is dominated. The lack of understanding of the process-microstructure-creep properties correlations, as well as the limited availability of creep properties data, has caused this technical breakthrough to be delayed. Due to the complexity of FGMs material, it is advisable to start with pure materials as a first step to understand the high-temperature mechanical properties for FGMs 316L SS-IN718 fabricated by DED. In the present study, the creep behavior of pure IN718 alloy fabricated by DED is investigated, while the creep properties for pure 316L SS fabricated by DED will be studied in another work.

The available investigations on creep performance for AM-IN718 mainly focus on L-PBF IN718 (L-PBF is the abbreviation of the powder bed fusion technique with laser as an energy source) [8,9,10,11,12,13,14,15,16,17,18,19,20]. The heat treatment and sample orientation were the primary parameters studied for the creep performance of L-PBF IN718 with comparison to their conventional counterparts. The results revealed that the creep behavior of L-PBF IN718 parts were generally inferior to the conventional processed counterparts at tension creep conditions. Microstructural features involving finer columnar grains, high dislocation density cellular substructures and defects, such as gas porosity and lack of fusion voids, were considered as the reason for the inferior creep property in the L-PBF IN718. It is well known that the cooling rate is significantly lower (~10^3^ k/s) in the L-DED process compared to the L-PBF process (10^5^–10^6^ k/s), Therefore, relatively coarser grain features are formed during the L-DED process, which is believed to be beneficial for the improvement of creep performance. In addition, the variety in microstructural characteristics in both AM techniques is expected to result in a distinct creep performance response even though the identical heat treatment is applied. There is, however, little information regarding the creep behavior of L-DED IN718. Thereby, it is crucial to investigate the influence of distinctive microstructures produced by L-DED on the heat treatment and the subsequent creep behavior of L-DED IN718.

The advantage of coarse microstructural characteristics produced by the L-DED process on the enhancement of creep strength and creep ductility has been firstly proposed in our previous work [21]. In that work, the creep deformation and fracture mechanism of a coarse-grained L-DED IN718 specimen subjected to the identical standard heat treatment as conventional wrought was investigated at 650 °C/750 MPa and compared to the conventional wrought specimen. Meanwhile, anisotropy in the creep behavior of L-DED IN718 was also studied. Our work showed that L-DED IN718 heat-treated under a lower temperature still preserved the cellular substructure, and columnar grains contributed to a superior creep strength relative to conventional wrought state and other heat-treated L-DED IN718. However, some questions are still open. First, the creep tests were short in duration and the difference in creep time for some investigated conditions was hardly distinguished, which made it difficult to clarify the primarily microstructural factors influencing creep strength for those conditions. Second, only standard heat treatments were applied in previous work. Heat treatment with a higher temperature is required in order to verify if the worst creep performance observed was caused by the partial recrystallization due to the lower heat treatment temperature. As we know, the creep damage is easily initiated and propagated along the interface between the fine and coarse grains [22].

Therefore, the aim of the present study is to continue to investigate the primary factor for the improvement of creep properties of L-DED IN718 with a coarse grain microstructure, and to develop an optimal heat treatment for it. In the present work, various heat treatments prevailing in L-PBF IN718 studies including homogenization heat treatment + double aging (HA-1080 °C, HA-1180 °C) and hot isostatic pressing + double aging (HIP+DA) treatments were applied to the as-deposited specimens as a first step to understand the influence of commonly observed heat treatment on the L-DED IN718 with coarse microstructural characteristics. Elaborate microstructure characterizations and fracture examinations were carried out in order to explain the effect of microstructural changes on the creep deformation and the crack mechanism.

## 2. Materials and Methods

### 2.1. L-DED IN718 Fabrication

IN718 blocks were fabricated by using a L-DED system (InssTek MX-600, Daejeon, Republic of Korea) in the current study. Commercial IN718 powder with a particle size ranging from 50 to 150 μm (AP&C, Saint-Eustache, QC, Canada) was used to manufacture IN718 samples. The main process parameters were as follows: laser power 373 W, laser beam size 0.8 mm, layer thickness 0.25 mm, hatch distance 0.5 mm, scanning speed 14 mm/s. A commercial wrought alloy was prepared in order to compare the creep behavior of L-DED samples to a traditionally processed sample. The L-DED-deposited IN718 blocks, while still attached on the substrate, were heat treated by various post process cycles. Heat treatments applied in the current work are summarized in Table 1.

### 2.2. Creep Testing

After various heat treatments, wire electro-discharge machining (WEDM) was used to cut creep specimens with a dimension of 2 mm × 1 mm × 10 mm. The tension loading direction during creep testing was perpendicular to the building direction (BD) [23]. Tension creep testing for all investigated cases were carried out at 650 °C and 700 MPa in standard lever arm creep machine. The creep elongation was recorded during the testing with use of an HBM WA/20 mm displacement transducer attached to the centerline rod connected to the specimen. The specimens were held for one hour before initiating the creep test. In order to obtain a reliable result, four tests were repeated for each tested case. For comparison, creep tests were also performed for samples heat treated by a simple double aging and benchmark material-wrought sample at the identical temperature and stress level.

### 2.3. Microstructure and Fracture Characterization

Standard metallography was used to prepare samples for microstructure characterization and examination. The polished samples were etched for a few seconds using a Glyceregia etchant (15 mL HCl, 10 mL glycerol and 5 mL HNO_3_). The microstructure of samples prior to creep testing and fracture mechanism of ruptured samples after creep testing were examined by using OM (optical microscopy; Nikon Eclipse MA200, Tokyo, Japan), SEM (scanning electron microscopy; JEOL IT 500 HR, JEOL Ltd., Tokyo, Japan), and EBSD (electron backscatter diffraction; EDAX Hikari Super camera, EDAX LLC, Mahwah, NJ, USA). A further observation of dislocation cell substructure was carried out by using a JOEL JEM-2100 transmission electron microscope (TEM) equipped with EDAX energy-dispersive X-ray spectroscopy (EDS) detector [21].

## 3. Results

### 3.1. Microstructure Evolution

Figure 1 depicts the inverse pole figure (IPF) maps for HA-1080 °C, HA-1180 °C and HIP-DA samples. As shown, recrystallization and grain coarsening occurred when the heat treatment temperature was above 1080 °C. Irregular columnar grains produced during L-DED process were transformed into uneven, coarser and equiaxed grains. Furthermore, complex network of high angle grains boundaries formed due to the irregular columnar grains were also transited to the relatively regular grain boundaries (GBs). The grain size slightly increased with the increasing temperature of the heat treatment. The average area-weighted grain size evaluated from EBSD data for all various heat-treated L-DED samples was 318 ± 161 μm (HA-1080 °C), 397 ± 193 μm (HA-1180 °C), 350 ± 151 μm (HIP-DA), respectively.

Figure 2 shows the evolution of the phase formed at various heat treatment conditions. In contrast to the long-striped Laves and needle-like δ phase found in the low temperature heat-treated L-DED samples in [21], a further increase in the temperature of heat treatment over 1080 °C contributed to a microstructure without Laves and δ phases. Only Nb and Ti-rich carbides were observed throughout the matrix of the HA-1080 °C sample in Figure 2a. The δ phase was absent at this temperature range since 1080 °C is above the solvus temperature of δ phase (1045 °C). Similar to the HA-1080 °C sample, only carbide rich in Nb and Ti was found throughout the matrix for HA-1180 °C in Figure 2b and HIP-DA samples in Figure 2c. Nevertheless, a higher temperature of heat treatment promoted the redistribution of Nb- and Ti-alloying elements in carbides back to the matrix and the microstructure was homogenized. In addition, the amount of carbides decreased and the size of carbide particles was slightly larger in the case of HA-1180 °C and HIP-DA samples compared to HA-1080 °C sample, due to the exposed higher temperature and longer duration time. Moreover, with the temperature of the heat treatment rising above 1080 °C, dislocation cell substructures observed in the as-deposited L-DED and DA samples completely disappeared. The measurement of the average length of the primary strengthening γ″ particles indicated that, HA-1080 °C, HA-1180 °C and HIP-DA L-DED samples had the similar size of primarily strengthening γ″ particles (20 ± 8 nm) as wrought sample, while slightly bigger γ″ particles in the DA sample (23 ± 6 nm).

### 3.2. Creep Tests

Figure 3 depicts the creep curves of all investigated cases at 650 °C/700 MPa. In addition to HA-1080 °C, HA-1180 °C and HIP-DA specimens, creep tests of L-DED sample heat treated by a simple double aging treatment (DA) and the conventional wrought sample were also carried out at the identical temperature and stress level. A summary of graphs for the creep life time t_r_, the minimum creep rate ε˙_min_, and the creep elongation at fracture ε_f_ are presented in Figure 4. The measurements of the creep results are listed in Table 2. Note that the creep result of as-deposited L-DED (AD L-DED) sample was not given here, due to the yield strength of the AD sample at 650 °C being much lower than the stress applied in the present study; thus, the sample was immediately fractured at the present stress. As shown in Figure 3 and Figure 4, the DA sample achieved the longest tr and ε_f_, but the slowest ε˙_min_ among other heat-treated L-DED variants and wrought specimen. In addition, with the increase in the temperature of the heat treatment, t_r_ decreased in a sequence of DA > wrought > HA-1080 °C > HA-1180 °C > HIP-DA. The ε˙_min_ decreased conversely as t_r_ increased. The change of ε_f_ for all samples followed this order: DA > wrought > HA-1080 °C ≈ HA-1180 °C ≈ HIP-DA.

### 3.3. Fracture Mechanism

The fracture surfaces of ruptured creep L-DED specimens are shown in Figure 5. Similar to fracture the surfaces of the ruptured sample at 650 °C/750 MPa in [21], the creep fracture mechanism at 650 °C/700 MPa was dominated by the intergranular crack mode. However, a significant difference in the creep crack mechanism was observed with various heat treatments. The DA sample exhibited a ductile-like fracture surface. Dimpled intergranular cracks with tear ridges were observed, as indicated by the red arrow and yellow arrows in Figure 5a, respectively. Similar fracture characteristics were also observed at the fracture surfaces of post-creep samples for L-PBF IN718 [8,12]. However, compared to the DA sample, samples with recrystallized grains, namely the HA-1080 °C, HA-1180 °C and HIP-DA samples, possessed more brittle-like fracture surfaces with a bright rock-candy appearance indicating a worse creep resistance, as can be seen in Figure 5b of the representative HIP-DA sample. In addition, large amounts of white precipitates were also found on the surface of cleavage planes, as shown in Figure 5b. The wrought sample demonstrated a similar cleavage fracture surface compared to the HIP-DA sample, as depicted in Figure 5c. Nevertheless, many consecutive slip bands, highlighted by white arrows in Figure 5c, were found on the bright crystallographic planes in the case of the wrought sample, which was absent in HIP-DA sample.

Figure 6 shows the cross section of the gauge section of ruptured creep samples. A typical creep failure mechanism involving micro-void coalescence and material separation perpendicular to the loading direction appeared to occur in all specimens. Compared to the creep damage observed at 650 °C/750 MPa in [21], the ruptured sample at a lower stress level in the present study presented more typical creep damage characteristics of cavity-induced coalescence instead of separation between precipitates and GBs. Different damage accumulation was found in various samples as heat treatment introduced various precipitates being voids initiators. In the case of the DA sample, a few visible cracks can be seen on the gauge section perpendicular to the loading direction. Cavities were observed in the vicinity of the white Laves phases, as highlighted by white arrows in Figure 6a. This implied that the intergranular cracks in the DA sample were mainly caused by the coalescence of Laves particles-induced micro-cavities at GBs. For the samples heat treated at the temperature above 1080 °C, with the absence of the Laves phase and δ phase, carbides at GBs may serve as the location for cavity nucleation when the grain boundary slides while creep testing, contributing to the intergranular crack mode. The longer cracks were formed mainly due to the accumulation of small r-type voids (forming at the GBs) around carbides, as highlighted in white arrows in Figure 6b of the representative HIP-DA sample. For the wrought sample, significant wedge type (w-type) voids, instead of visible cracks present in the L-DED samples, were found at the tripe junctions of GBs. The creep failure mechanism for the wrought sample was likely to be caused by the accumulation of w-type voids due to the weak interface between δ phase and the GBs marked in Figure 6c.

Figure 7 shows the representative IPF maps and KAM (Kernal Average Misorientation) maps in order to clarify the creep deformation mechanism. A significant plastic deformation was seen near the creep fracture surface of DA specimen in Figure 7a, which was indicated by a higher KAM shown in Figure 7b. However, the HIP-DA and wrought samples with the absence of a L-DED-produced microstructure showed little plastic deformation with a smaller KAM after the creep fracture, as shown in Figure 7d and Figure 7f, respectively. Moreover, local plastic deformation with a higher KAM near GBs in all tested samples clearly illustrated that cracks were initiated and propagated along the interface between the secondary phase particles (Laves, carbide and δ phases) and the GBs.

## 4. Discussion

To explain the differences in the creep deformation and fracture mechanism for L-DED specimens, various microstructural factors influencing the creep properties will be evaluated in this section.

The DA sample exhibited a superior creep life and creep ductility simultaneously compared to the conventional wrought sample and other heat-treated L-DED samples. This improvement in creep strength and ductility can be attributed to the reserved dislocation cell substructure from the L-DED process, which is considered the major factor dominating the creep deformation mechanism, particularly during the primary and secondary creep stages. This is due to the dislocation cell substructure is the primary microstructural feature that differs significantly from the DA sample to other heat-treated L-DED variants and the conventional wrought sample. Previous work [24,25,26,27] reported that tensile strength and ductility at room temperature can be enhanced simultaneously by the dislocation cell structure, due to the superior ability in dislocation storage upon deformation. It is well known, however, that the high room temperature strength of materials is generally unfavorable for the creep performances, which is supported by the fact that a superior room temperature strength can be easily achieved. In contrast, a worse creep performance of L-PBF materials was produced when compared to the wrought sample. High tensile strength enhanced by a high dislocation density-aided cell substructure impedes the relaxation of stress concentration that has built up around the secondary phases, facilitating the formation and coalescence of creep cavities [13].

However, in the present study, the coarse dislocation cell substructure of DA sample can not only improve the room temperature tensile strength but also contribute to a superior creep strength and creep ductility. This discrepancy of the role of dislocation cell substructures on creep behavior of AM-IN718 is probably associated with the dislocation density and the dislocation cell size. For example, a mean cell size of 0.6 μm was reported in the case of as-printed L-PBF IN718 sample, whereas the average cell size was 5 μm, nearly 10 times bigger, in the case of the present as-deposited L-DED IN718 sample. Though it is challenging to quantitatively determine the dislocation density in such coarse grains with a much bigger cell substructure size, one thing is clear: the dislocation density surrounding the cell wall in the L-DED IN718 sample is lower than that of L-PBF IN718 due to the lower cooling rate. As the high density dislocation can stabilize the thermal stability of cell substructures during heat treatment [27], the cell substructure with a lower dislocation density for L-DED IN718 diminished at a lower solution temperature, i.e., 980 °C, where cell substructure still remained for L-PBF IN718 in previous reports [18,28]. This variety in the cell substructure between L-DED and L-PBF would lead to a different creep performance response even though identical heat treatment was applied for both microstructures. As proposed by Calderon et al. [29], the dislocation cell structure is the main contributor to the shorter rupture times, particularly for the primary creep stage when studying the creep behavior of L-PBF-316L SS. They stated that the high-density cell substructure can limit the strain hardening capacity and, thus, result in a shorter primary creep duration in the L-PBF compared to the conventional 316L SS. Therefore, it is reasonable to argue that the L-DED DA sample, which partly diminishes the initial dislocation density but still preserves the cell substructure, is beneficial for the improvement of the creep properties of IN718 via L-DED. However, apart from the dislocation cell substructure, other microstructural factors such as precipitation, grain size and various intermediate phases may also interact with the dislocation cell structure, which mutually influence the creep behavior of L-DED IN718. Thereby, the relationship between the dislocation cell structure and the creep mechanism of AM-IN718 still needs to be examined carefully in the future in order to clarify the impact on creep performance for AM-IN718.

Another noteworthy result is that the creep life decreased with the increase in heat treatment temperature, while the steady creep rate increased conversely. To explain this trend, microstructural features, i.e., precipitates and grain size, will be examined. As reported in previous studies [30,31], at the applied temperature and stress in the current study, the dislocation power law mechanism dominates the creep performance. The intermediate phases in IN718, Laves and δ phases, which consume a high content of Nb for the precipitation of hardening γ′/γ″ phases, deteriorate the creep properties. Thereby, increasing heat treatment temperature is expected to lead to a better creep performance, as a higher heat treatment temperature can dissolve Laves and δ phases and release Nb back to the matrix for precipitating γ′/γ″ particles. However, the samples heat treated above 1080 °C did not show an increase in creep life with comparison to the DA sample. It implies that the improvement of creep performance due to the increased available amount of Nb is smaller than the deterioration due to the loss of dislocation cell substructures strengthening.

Increase in grain size may also contribute to worse creep performances for samples with recrystallized grains when compared to the DA sample. Figure 8 demonstrates the creep life of various investigated samples dependent on grain size. As shown, larger grains are not beneficial for the superior creep behavior in the current study. The majority of heat-treated L-DED specimens with significantly coarser grains achieved much worse creep life compared to the wrought specimen. Samples heat treated at temperature above 1080 °C showed a remarkable deterioration in creep life compared to the wrought variant, which can be attributed to the loss of L-DED-produced microstructural characteristics and the occurrence of grain coarsening. In the dislocation creep mechanism, fine grains are considered to contribute to better creep properties due to the grain boundaries strengthening mechanism by hindering the dislocation motion, which was supported by the study of Xu et al. [20]. Fine grains in the wrought sample show a good rotation ability and yield more deformation, confirmed by the evidence of cleavage planes decorated with extensive slip bands, as shown in Figure 5c. Furthermore, much finer equiaxed grains in the wrought sample help to slow down the crack propagation by altering the crack growth direction, as marked in white arrows in Figure 7f However, much coarser grains in recrystallized samples, HA-1080 °C, HA-1180 °C and HIP-DA with regular GBs shown in Figure 7c,d, may help cracks grow and result in worse creep properties. That may explain why, despite the absence of dislocation cell substructure and significantly undesirable δ phases at GBs in the wrought sample, the wrought sample nevertheless shows a superior creep performance to other recrystallized samples.

Secondary phase, carbides will be considered in order to understand the differences in creep life for HA-1080 °C, HA-1180 °C and HIP-DA samples and their worse creep ductility. The present work, along with other studies [11,20,32,33], has demonstrated that the Laves and δ phases at GBs are undesirable for the creep performance since they can serve as crack initiation sites during creep process. Apart from Laves and δ phases, closely spaced NbC particles have also shown to lead to intergranular fracture, which is consistent with the results of Sundaraman et al. [34] and McLouth et al. [16]. The size, number and distribution of these secondary phases significantly impact the creep deformation and failure mechanism [33]. Therefore, the worse ductility for those recrystallized samples in the present study may be due to the carbides rich in Nb, Ti precipitated at grain boundaries resulting in the weakening of GBs and thus causing intergranular cracks. A slight increase in size of carbides particles may explain the trend of decreased creep life in an order of HA-1080 °C, HA-1180 °C and HIP-DA samples. Note that the HIP-DA heat treatment applied in the present study did not effectively improve the creep properties of L-DED IN718 samples compared to L-PBF IN718 [12]. This might be due to the lower number of pores formed during the L-DED process compared to the L-PBF process. The porosity was 0.01% in the as-deposited L-DED sample and was reduced to 0.005% after HIP. Since the major purpose of HIP is to improve microstructural characteristics by effectively closing porosity, the application of HIP will cause less impact on the denser L-DED IN718 alloy.

In summary, the reserved dislocation cell substructure from the L-DED process contributes to the enhanced creep strength and creep ductility in the present L-DED heat-treated samples. Smaller grain size is beneficial for an improvement of creep performance during the dislocation creep. The carbide secondary phase also affects the creep property, particularly the creep ductility. L-DED-induced defects are considered to have little effect on creep behavior and crack mechanism.

## 5. Conclusions

In this study, the creep behavior for a coarse-grained L-DED IN718 was investigated at 650 °C/700 MPa and compared to its wrought counterpart. The main findings can be concluded as follows. The HA-1080 °C + DA, HA-1180 °C + DA and HIP-DA heat treatments applied in the present study completely eliminate the L-DED produced microstructure, particularly the dislocation cell substructure. For those L-DED samples heat treated at a higher temperature, recrystallized, equiaxed grains are observed, and only carbides were found throughout the matrix. The loss of dislocation cell substructure and the coarsening of grains contributes to a poor creep performance compared to the conventional wrought sample and the DA sample. A simple double aging heat treatment, which can preserve the unique L-DED process-induced microstructural characteristics and promote precipitate strengthening γ′/γ″ phases, is optimal for the improvement of creep performance in the coarse-grained L-DED IN718. HIP heat treatment does not have a beneficial effect for the enhancement of creep properties in L-DED IN718.

## Figures and Tables

**Figure 1 materials-16-01377-f001:**
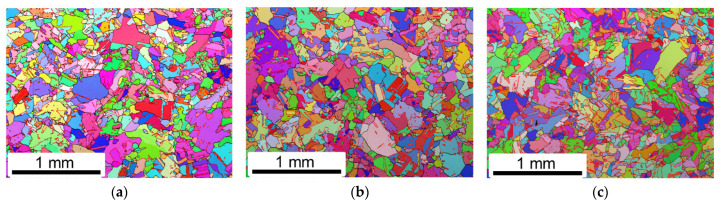
IPF maps: (**a**) HA-1080 °C, (**b**) HA-1180 °C and (**c**) HIP-DA.

**Figure 2 materials-16-01377-f002:**
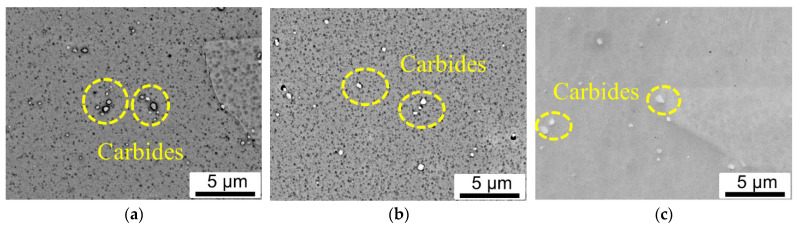
SEM-BSE (Back Scatter Electron) micrographs showing the phase evolution: (**a**) HA-1080 °C, (**b**) HA-1180 °C and (**c**) HIP-DA.

**Figure 3 materials-16-01377-f003:**
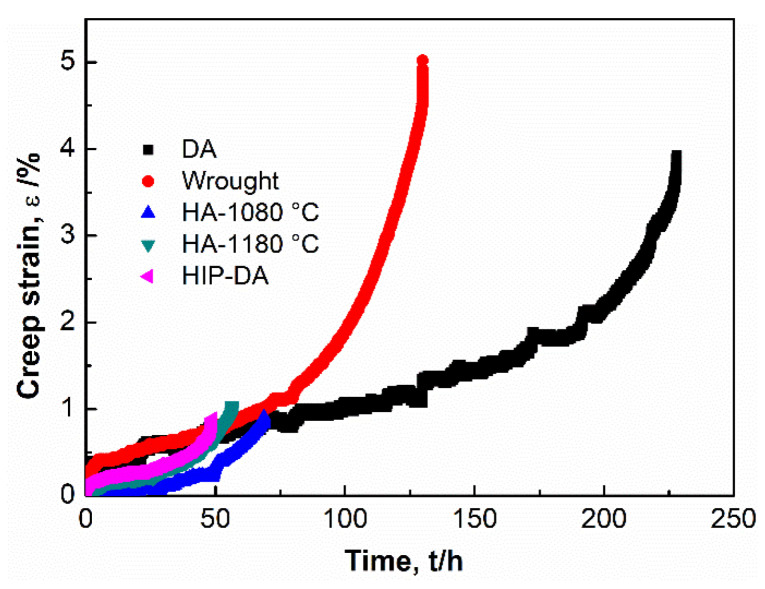
Creep lifetime-creep strain curves of L-DED heat-treated variants compared to the conventional wrought sample at 650 °C/700 MPa.

**Figure 4 materials-16-01377-f004:**
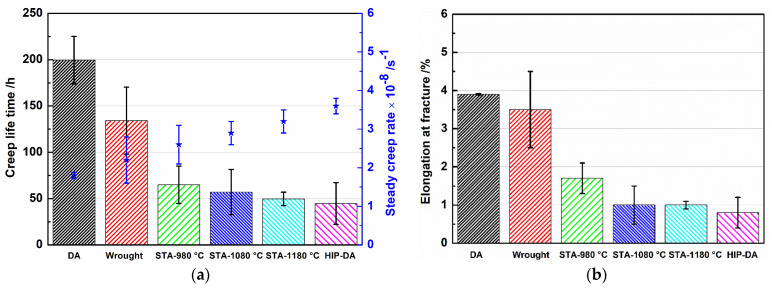
Graphs of the t_r_ and the ε˙_min_ (**a**) and the ε_f_ (**b**) for all investigated specimens.

**Figure 5 materials-16-01377-f005:**
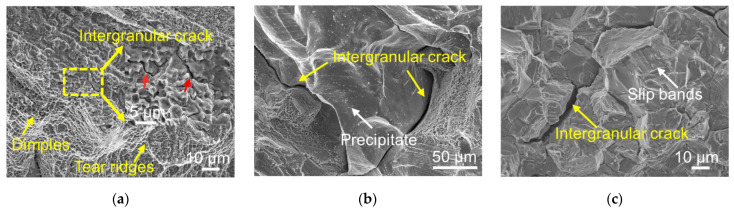
Fracture surfaces of post-creep samples tested at 650 °C/700 MPa: (**a**) DA, (**b**) HIP-DA, (**c**) Wrought.

**Figure 6 materials-16-01377-f006:**
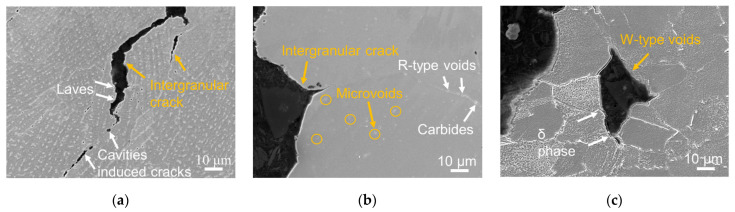
Ruptured creep specimens at 650 °C/700 MPa: (**a**) DA, (**b**) HIP-DA, (**c**) Wrought.

**Figure 7 materials-16-01377-f007:**
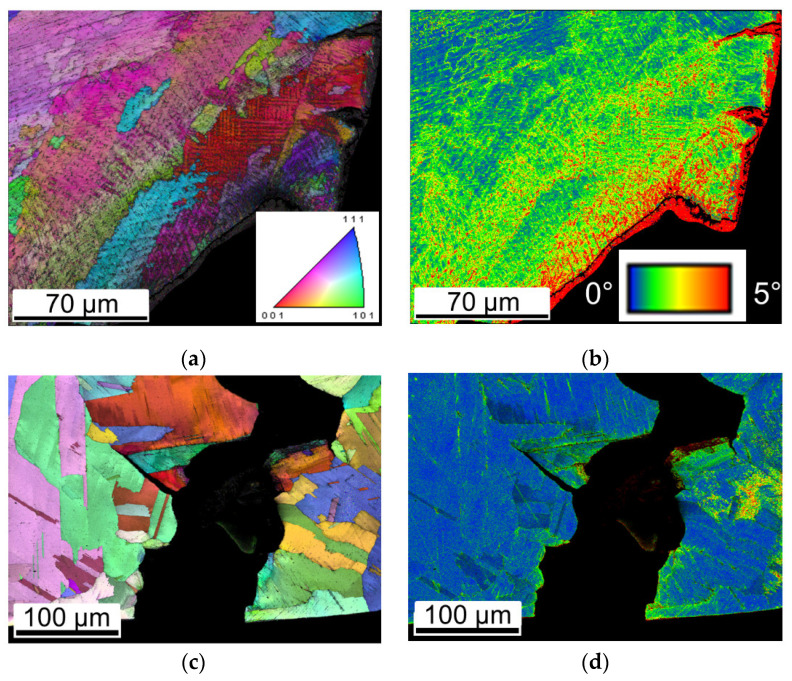
IPF maps of ruptured samples (**a**) DA, (**c**) HIP-DA, (**e**) Wrought; KAM maps (**b**) DA, (**d**) HIP-DA, and (**f**) Wrought, respectively.

**Figure 8 materials-16-01377-f008:**
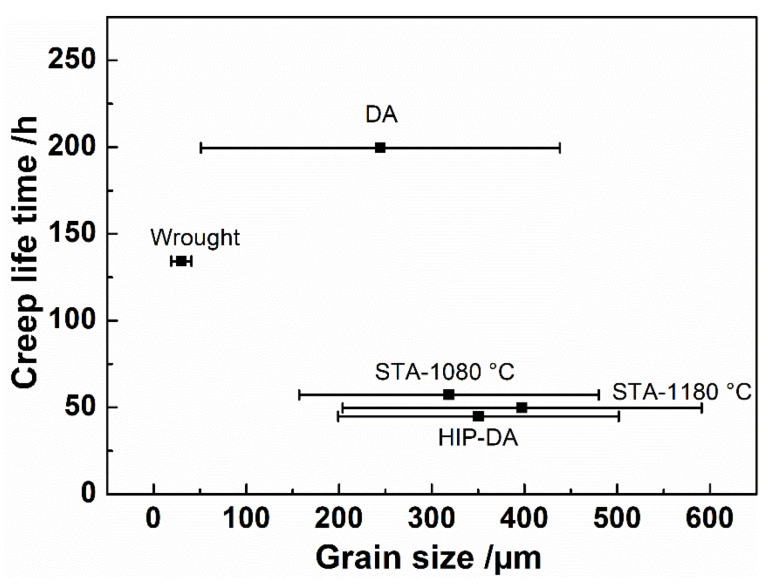
The creep lifetime dependence of the grain size measured for all investigated cases.

**Table 1 materials-16-01377-t001:** Heat treatment applied in the current work.

Condition	Designation	1st Step: Homogenization	2nd Step: Aging Hardening
Homogenization heat treatment at 1080 °C	HA-1080 °C	1080 °C/1 h/water cooling	720 °C/8 h/furnace cooling 50 °C/h + 620 °C/8 h/air cooling
Homogenization heat treatment at 1180 °C	HA-1180 °C	1180 °C/1 h/water cooling	
Hot isostatic pressing and double aging treatment	HIP-DA	1180 °C/100 MPa/4 h	

**Table 2 materials-16-01377-t002:** Summary of creep results for the t_r_, the ε˙_min_ and the ε_f_ for all tested cases.

	DA	Wrought	HA-1080 °C	HA-1180 °C	HIP-DA
t_r_/h	200 ± 26	134 ± 36	57 ± 24	50 ± 7	45 ± 23
ε˙_min_ × 10^−8^/s^−1^	1.8 ± 0.1	2.2 ± 0.6	2.9 ± 0.3	3.2 ± 0.3	3.6 ± 0.2
ε_f_/%	3.9 ± 0.1	3.5 ± 1.5	1.0 ± 0.5	1.0 ± 0.1	0.8 ± 0.4

## Data Availability

Data available in a publicly accessible repository.

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
