# Peer review of "Effect of Heat Treatment on Creep Deformation and Fracture Properties for a Coarse-Grained Inconel 718 Manufactured by Directed Energy Deposition"

_materials, 2023, doi:10.3390/ma16041377_

Round 1
Reviewer 1 Report
“Effect of heat treatment on creep deformation and fracture
properties for a coarse-grained Inconel 718 manufactured by directed energy deposition “investigates on creep behaviour of Laser Directed Energy Deposition of Inconel 718 alloy blocks and the effects that heat treatments have on their microstructural characteristics.
Revision or comment
The work is in interesting and cover process and materials that are more and more used in research and application with the aim to evaluate the best heat treatment to improve structural and morphological characteristics of the produced material.
More attention could be place to the introduction because more clear exposure would improve the comprehension of what it has been done by others and what is presented in the paper.
The decrease of the creep life time in figure 4 is not so evident by considering the standard deviation in the plot.
Authors are invited to check the paper for English language because some sentences are of difficult reading like the following examples:
Line 14: check the sentence “by subjected to various post heat treatments”, it should be clearer: “to be subjected to” or “and subjected to”
Line 74: replace “valid” with “verify”
Line 77: replace “clarify” with “investigate on”
…
Author Response
1. More attention could be place to the introduction because more clear exposure would improve the comprehension of what it has been done by others and what is presented in the paper.
Answer: Thank the reviewer for pointing out this valuable question. The investigation on creep properties of IN718 fabricated by L-DED technique and the heat treatment effect was done in our previous work [21] and the present work. The motivation was aroused in order to address the question: the literature available for creep behavior of AM-IN718 were only for L-PBF IN718. And their results showed that L-PBF IN718 had worse creep properties compared to conventional counterparts due to the fine microstructural characteristics involving fine grains, fine dislocation cells and high density dislocations, etc…… Then the idea came to us naturally that L-DED produced IN718 probably can improve the creep performance compared to L-PBF IN718. Since cooling rate and solidification conditions are significantly different in both two AM techniques, distinct microstructural features can be expected to form in both two AM techniques and thus the post heat treatment response. However, little or no literature can be found about the investigation on the coarse grain microstructure on the heat treatment effect and creep properties of L-DED IN718. Therefore, the objective of the present study is to study the L-DED- produced microstructure effect on heat treatment and subsequent creep behavior. Since more details about the motivation have already been explained in our previous study, the current study as the continuation of our previous work is mainly to emphasize the effect of higher temperature of heat treatment and compared to previous work. However, according to the reviewer suggestion, introduction about what new we have done has been added. Please check line 62-64.
2, The decrease of the creep life time in figure 4 is not so evident by considering the standard deviation in the plot.
Answer: We agree with the reviewer. As shown in figure 4, a quite large deviation can be found in the creep results which may be due to several factors. First, inhomogeneous grains consisting of small and large grains were formed even after high temperature of heat treatment in figure 1. Second, small creep sample size compared to grain size enabled the creep sample with different number and size of grains, which contributed to a large deviation. In order to have a reliable result, four specimens, which were cut from different locations of L-DED IN718 sample, were repeated for each investigated cases. We still can see this decrease in creep life time with increasing heat treatment temperature, even though not so evident. In addition, a striking decrease in creep life time was occurred at the heat treatment of 1080 °C which already eliminated all microstructural characteristics produced by L-DED. Undoubtedly, a further increasing heat treatment temperature cannot improve the creep properties. Thereby, the conclusion is quite clear that L-DED produced coarse grain structure, particularly the coarse dislocation cell substructure, is the major reason for improving the creep performance compared to L-PBF.
3.Authors are invited to check the paper for English language because some sentences are of difficult reading like the following examples:
Answer: Thank the reviewer for the suggestions, the whole manuscript was double checked and the revisions can be seen in red highlight.
Reviewer 2 Report
The paper deals with an interesting topic. The work is interesting, and the problem is analyzed in detail. Congratulations on a successful paper.
Author Response
We appreciate the positive comments about our work.
Reviewer 3 Report
The effect of heat treatment on creep deformation and fracture properties of coarse-grained for a coarse-grained Inconel 718 manufactured by directed energy deposition was investigated in this article. In general, the article is adequate and reliable, the workload is appropriate, and the discussion part is satisfying. However, some minor issues appearing in the article still need to be revised before acceptance for publication:
1. Regarding the description in lines 91 and 116 that "more details can be found in the previous paper", why not just describe it for the reader to read? This must be improved.
2. The naming of figures in the article needs to be consistent with the naming of figures cited in the main text, which need to be named "Figure" as required by the journal instead of the abbreviation "Fig" .
3.The error bar in Figure 4 needs to be replaced with a thicker line
4.The time axis of Figure 3 should reduce and the naming of the curve needs to move to the proper position accordingly to perhaps look fuller.
Author Response
Thank the reviewer for the useful and constructive suggestions for our work.
- We apologize for the inconvenience that caused during reading. More details about the methods and metallography were deleted due to the similarity check since almost same printing process was utilized as previous paper. Details in materials and methods were added in the new manuscript. Please have a look in the new Materials and methods.
-
Thank the reviewer for this suggestion. All Fig in the manuscript were revised to Figure.
-
Revised, the error bar has a thicker line in the revised version.
-
Figure 3 was modified in the revised version. please have a look.